# Expression Analysis of mRNA Decay of Maternal Genes during *Bombyx mori* Maternal-to-Zygotic Transition

**DOI:** 10.3390/ijms20225651

**Published:** 2019-11-12

**Authors:** Meirong Zhang, Pingzhen Xu, Huilin Pang, Tao Chen, Guozheng Zhang

**Affiliations:** 1School of Biotechnology, Jiangsu University of Science and Technology, Sibaidu Rd, Zhenjiang 212018, China; 2Sericulture Research Institute, Chinese Academy of Agricultural Sciences, Sibaidu Rd, Zhenjiang 212018, China

**Keywords:** *Bombyx mori*, maternal gene, decay, embryonic development, maternal-to zygotic-transition

## Abstract

Maternal genes play an important role in the early embryonic development of the silkworm. Early embryonic development without new transcription depends on maternal components stored in the egg during oocyte maturation. The maternal-to-zygotic transition (MZT) is a tightly regulated process that includes maternal mRNAs elimination and zygotic transcription initiation. This process has been extensively studied within model species. Each model organism has a unique pattern of maternal transcriptional clearance classes in MZT. In this study, we identified 66 maternal genes through bioinformatics analysis and expression analysis in the eggs of silkworm virgin moths (*Bombyx mori*). All 66 maternal genes were expressed in vitellogenesis in day eight female pupae. During MZT, the degradation of maternal gene mRNAs could be divided into three clusters. We found that eight maternal genes of cluster 1 remained stable from 0 to 3.0 h, 17 maternal genes of cluster 2 were significantly decayed from 0.5 to 1.0 h and 41 maternal genes of cluster 3 were significantly decayed after 1.5 h. Therefore, the initial time-point of degradation of cluster 2 was earlier than that of cluster 3. The maternal gene mRNAs decay of clusters 2 and 3 is first initiated by maternal degradation activity. Our study expands upon the identification of silkworm maternal genes and provides a perspective for further research of the embryo development in *Bombyx mori*.

## 1. Introduction

The transition from the oocyte depending on maternally supplied RNA and protein complements the commencement of zygotic transcription is a key process in the earliest stages of early embryonic development [1,2,3]. Early embryonic development is maternally regulated. Maternal mRNAs and proteins stored in oocytes are activated to initiate and regulate embryonic development. Following the period of maternal transcriptional silence, the embryonic zygote’s own genome starts transcription and plays a role in the development of embryos [4]. The transition from the maternal to the zygotic genome is a key process in the final transformation of the zygotic regulation of individual development [2]. Following the maternal-to-zygotic transition (MZT) period, the maternal control of development begins to decline and maternal mRNAs begin to degrade [5,6]. Therefore, later developmental control is exhibited via a combination of the maternal RNAs and proteins being eliminated and the zygotic genome becoming transcriptionally active [7].

The elimination of these maternal mRNAs is completed through two kinds of activities: Maternal-source encoding (maternal degradation activity) and zygotic transcription (zygotic degradation activity) [7,8]. Some molecular mechanisms regulating maternal mRNA clearance have been previously demonstrated. RNA-binding proteins (RBPs) play an important role in directing the decay of maternal mRNAs in *Drosophila*. Smaug (SMG) RBP participates in clearance of maternal mRNAs via binding maternal transcripts that contain SMG recognition cis-elements (SREs) [2,9,10,11,12,13]. An additional cis-element, such as Pumilio-like binding element (PBE), was also identified and is bound by Pumilio (PUM) RBP, a post-transcriptional regulator implicated in both translational repression and the destabilization of a specific subset of maternal mRNAs [12,14,15,16,17,18]. PUM has been shown to interact with brain tumor (BRAT) RBP [19]. BRAT can directly bind to RNA and mediate the decay of maternal mRNAs [20,21]. BRAT and SMG can recruit and/or stabilize ME31B (RNA-binding protein) on their targets in maternal mRNA clearance [22]. ME31B exists in complexes that also contain eIF4E (binding 5’ cap), Cup, Trailer Hitch (TRAL), and polyadenylate binding protein (PABP) (binding 3’ poly (A) tail) [22,23]. PIWI-associated RNAs (piRNAs) and their associated proteins act together with SMG to recruit the deadenylase CCR4 deadenylation complex to *Nanos* maternal mRNA, thus promoting its decay during early embryogenesis in *Drosophila* [24,25]. In *Drosophila*, the RNA-binding proteins of SMG, BRAT, and PUM bind to and direct the degradation of largely distinct subsets of maternal mRNAs in both maternal and zygotic degradation activities [17,21,22,26]. SMG is also essential for the synthesis of microRNAs (miRNAs) during the *Drosophila* maternal-to-zygotic transition [26,27,28]. miRNAs have important functions during early embryonic development in metazoans [29,30]. miRNAs facilitate the transition from an oocyte-inherited to an embryonic transcriptome by eliminating maternal mRNAs during MZT in *Drosophila*, zebrafish (*Danio rerio*), and *Xenopus* [5,31,32,33]. Codon identity regulates the maternal program of mRNA decay, and codon composition shapes maternal mRNA clearance during the maternal-to-zygotic transition in zebrafish, *Xenopus*, mouse (*Mus musculus*), and *Drosophila* [34,35,36]. Codon-mediated decay and miRNAs induced decay evolutionarily conserved mechanisms for modulating mRNA stability in metazoans [30,31,34,35].

The joint action of maternal and zygotic degradation signaling pathways triggers the clearance mechanism of maternal components both in temporal and spatial axes [8]. The biological functions of maternal mRNA elimination during MZT remain unclear thus far. However, the potential functions of this process can be hypothesized [2,7]. In *Drosophila*, maternal mRNA degradation starts soon after egg activation and is largely complete by the third hour of embryogenesis [8,11,12]. During the early embryo stage, maternal transcript clearance may play a passive role [7]. Permissive functions may be necessary to allow newly synthesized zygotic transcripts to exert their functions [18,37,38,39,40], whereas instructive functions regulate developmental progress [41,42].

The embryonic development of *Bombyx mori* is significantly different from that of *Drosophila*. The progress of egg formation in different positions of the ovariole is inconsistent. According to various morphological criteria, the development of the follicles is divided into 12 different stages [43]. During vitellogenesis (stages 4–10), the oocyte increases gradually in volume and is filled with yolk spheres, lipid droplets, and glycogen granules. At the end of this period, degenerated nurse cells are devoured by follicular epithelial cells [43]. In the choriogenesis period (stages 11, 12), different types of eggshell proteins are synthesized and secreted successively to construct the eggshell. The developmental stages of each ovariole are opportune, found in vitellogenesis, choriogenesis, and mature eggs from day 8 pupae [43,44]. Following the choriogenesis period, egg maturation occurs [43,44]. The time of sperm entering the egg occurs a few seconds before the egg leaves the mother. The union of sperm and egg pronuclei occurs at about two hours after silkworm eggs are laid [45,46,47].

In our previous study, we identified 76 potential maternal genes in silkworm via orthologous comparison [48]. In this study, further sequence alignment analysis and the identification of these potential maternal genes were performed. Expression patterns were analyzed in eggs of virgin moths to identify the maternal genes. In this study, the expression of the 66 successfully identified maternal genes was analyzed in the developing oocytes from day eight female pupae, and during the MZT period in silkworm.

## 2. Results

### 2.1. Identification of Potential Maternal Genes

In our previous study, we obtained 76 potential maternal genes in the *B*. *mori* genome [48]. In this study, we blasted the NCBI database and the newly assembled silkBase by the sequence of each gene that was obtained from the silkworm database (SilkDB) [49]. BGIBMGA012517 and BGIBMGA012518 are orthologous genes to *MAMO* in *Drosophila melanogaster* [48]. The sequences of BGIBMGA012517 and BGIBMGA012518 were found to be part of the KWMTBOMO05086 gene that was annotated in silkBase (Table 1). Similarly, BGIBMGA002518and 002519 were found to be part of KWMTBOMO005319; BGIBMGA000972, 000973, and 000974 were part of KWMTBOMO007913; BGIBMGA004415 and 004416 were part of XM_012695102; and BGIBMGA013473 and 013474 were part of XM_012690736. The BLAST results of BGIBMGA007314 and BGIBMGA001094 were very poor in the NCBI database and silkBase. Therefore, 68 preliminary potential maternal genes were identified in the *B. mori* genome. The mRNAs of maternal genes are produced by the females and stored in embryos [2,6]. Thus, undetectable expression in the embryo can be considered a non-maternal gene. The results of the transcriptional analysis of the 68 potential maternal genes in eggs of virgin moths by reverse transcription-PCR (RT-PCR) showed that for 66 genes, transcriptional signals were detected, whereas two had no transcriptional signals (Figure 1), BGIBMGA003296 and BGIBMGA002069 had no transcriptional signals (Figure 1). The specific primers for each gene were used in RT-PCR, as shown in Appendix A. We finally identified 66 maternal genes in the silkworm genome, and information, including amino-acid length, chromosomal distribution, signal peptide, and gene name, was collected for each (Table 1).

### 2.2. Tissue Expression Patterns on Day 3 of the Fifth Instar

The silkworm feeds and grows quickly in the fifth larval period. Day 3 of the fifth instar is typical for larval development with more active biological processes [50]. Therefore, studying this time point will enrich the expression patterns and help with further understanding of the functions of maternal genes in different developmental stages. The microarray data of 10 silkworm tissues on day 3 of the fifth instar were downloaded from the SilkMDB [50]. The probes of *SPE*, *BAEE* and *Pabn2* were not found in SilkMDB from the attached BLAST search (Table 1). The microarray data of the other 63 maternal genes are provided in Appendix A. The expression patterns of these 63 maternal genes are listed as found in various tissues and both sexes of silkworm in Figure 2. The expressed genes are defined as previously described [51]. Most of the maternal genes usually showed very low expression levels overall in the tissues and sex. The expression levels of sw10899 (*aub*) and sw14777 (*me31B*) were higher in the ovary and testis than in other tissues overall, sw19434 (*Nelf-E*) was only higher in the testis. The expression level of sw20327 (*proPPAE*) was higher in the testis, head, epidermis, and hemocyte, and sw13482 (*Th*) was higher in the head and epidermis. The expression levels of sw22934 (*Eif-4a*), sw12663 (*eIF4AIII*), sw1118 (*Bin1*), and sw21871 (*Sod2*) were higher overall. Most maternal genes showed low expression levels in multiple silkworm larval tissues on day 3 of the fifth instar. This is contrary to the abundant expression in the eggs of virgin moth (Figure 1).

### 2.3. Expression Analysis in Developing Oocytes in Day 8 Pupae

The silkworm has a pair of ovaries, each of which is composed of four ovarioles. The developmental stage of each ovariole is opportune, being found in vitellogenesis, choriogenesis, and mature eggs from day 8 pupae [43,44]. A large amount of yolk proteins and no chorion proteins exist in oocytes during vitellogenesis. Chorion proteins appear just after vitellogenesis and continue throughout the whole of choriogenesis and until the formation of the eggshell of mature eggs [43,44,52,53,54]. The expression of the 68 potential maternal genes in the vitellogenesis, choriogenesis, and mature eggs in day 8 pupae was analyzed by RT-PCR. The result showed that for 66 genes, transcriptional signals were detected, whereas BGIBMGA003296 and BGIBMGA002069 also had no transcriptional signals (Figure 3). *Me31B* and the other 31 genes (in total 32) showed consistent expression levels in the vitellogenesis, choriogenesis, and mature eggs in day 8 pupae (Figure 3). *Hip14* (*ZDHHC17*) and the other 33 genes (in total 34) had transcriptional signals and presented different trends in expression in the vitellogenesis, choriogenesis, and mature eggs in day 8 pupae (Figure 3).

### 2.4. Transcriptional Degradation during the Maternal-to-Zygotic Transition

To identify the transcriptional degradation patterns of the 66 maternal genes during different developmental stage embryos, six time-series samples were collected at 0, 0.5, 1.0, 1.5, 2.0, and 3.0 h after fertilized embryo spawning, and were analyzed by reverse transcription-quantitative PCR (RT-qPCR). The specific primers for each gene that was subjected to RT-qPCR are shown in Appendix A. In total, temporal control of their transcript clearance presented three different maternal transcript clusters during the maternal-to-zygotic transition (Figure 4, Figure 5 and Figure 6, Table 2).

In cluster 1 (Table 2), the transcript levels of 8 maternal genes (*Sod2*, *Eif-4a*, *eIF4AIII*, *bai*, *Pabn2*, *Bin1*, *Chc* and *tud*) showed no change from 0 to 3 h (Figure 4). The tissue expression levels of *Eif-4a* (sw22934), *eIF4AIII* (sw12663), *Bin1* (sw1118), and *Sod2* (sw21871) were high and uniform overall in 10 tissues at day 3 of the silkworm fifth instar (Figure 2). For a closer examination, we used RT-PCR to investigate these eight maternal genes and their transcript temporal control from 0 to 18 h after fertilized embryo spawning (Appendix A). The transcripts of *Sod2*, *Eif-4a*, *eIF4AIII*, *Bin1*, *Chc*, and *tud* kept consistent levels from 0 to 18 h after fertilized embryo spawning, respectively (Appendix A). *Pabn2* and *bai* presented changing trends in expression at the transcriptional level (Appendix A).

Regarding the other 58 maternal genes, the RT-qPCR results showed that their transcripts significantly changed with two main characteristics during the maternal-to-zygotic transition (MZT). In cluster 2 (Table 2), the transcripts of 17 genes were significantly decreased from 0.5 to 3.0 h (Figure 5). In cluster 3 (Table 2), the transcripts of 41 genes were decreased significantly after 1.5 h (Figure 6). This indicates that the maternally supplied mRNAs of most maternal genes were universally degraded during MZT. Unlike other genes, the transcript of the *wbl* gene was decreased significantly from 0.5 to 2.0 h, and increased sharply at 3.0 h. This transcript belongs to cluster 2 and is an exception.

## 3. Discussion

In our previous study, 76 potential silkworm maternal genes were identified by orthologous comparison [48]. In this study, 68 of the 76 potential silkworm maternal genes were initially identified through further sequence alignment analysis, and 2 of the 68 maternal genes were not expressed in the silkworm eggs of virgin moths. The mRNAs of maternal genes are produced by females and loaded into the embryos [2,6]. Thus, the expression of a gene was not detected in embryos that can be identified as a non-maternal gene. A total of 66 maternal genes were finally identified in silkworm.

The embryonic development of *B. mori* is significantly different from that of *Drosophila*. The silkworm has a pair of ovaries each composed of four ovarioles, each of which contains a chain of follicles [43,52]. The previous research on in vitro culturing of *B. mori* ovarian follicles showed that follicle development starts from middle vitellogenesis to late choriogenesis [55]. The follicles develop depending on an endogenous developmental program that does not require the presence of additional factors from tissues outside the ovariole [43,52]. Each follicle is composed of an oocyte and seven nurse cells surrounded by a single layer of the follicular epithelium [43]. The degenerated nurse cells are devoured by follicular epithelial cells at the end of vitellogenesis [43]. The 66 maternal genes were expressed in vitellogenesis on day 8 female pupae, which suggests that maternal mRNA is derived from the nurse cells.

The development of silkworm follicles is divided into 12 different stages [43]. The developmental stage of each ovariole is opportune, being found in vitellogenesis, choriogenesis, and mature eggs from day 8 pupae [43,44]. The rate of progression of vitellogenesis toward choriogenesis is estimated to be 2–2.5 h per follicle [56,57]. The eggs are considered mature upon finishing the formation of the eggshell in the choriogenesis period, and the mature eggs first appear in each ovariole proximal oviduct in day 8 female pupae [43,44,52]. The maternal genes of *Hip14* (*ZDHHC17*) and the other 26 genes (27 in total) had higher expression levels in vitellogenesis than in choriogenesis and mature eggs, whereas their expression levels were similar in choriogenesis and mature eggs on day 8 female pupae. These 27 maternal genes may have biological functions in the developmental process from vitellogenesis to choriogenesis in *B. mori*.

The MZT is a tightly regulated process that is identified by the elimination of maternal mRNAs and the initiation of zygotic transcription. This process has been extensively studied within model species. Each model organism has a unique pattern of maternal transcriptional clearance classes during the MZT. Four subsets of transcripts were characterized in *Drosophila*: Stable mRNAs, mRNAs targeted solely by the maternal or the zygotic degradation pathway, and those targeted by both pathways [7,12,31,58]. In activated, unfertilized eggs of *Drosophila*, maternal decay activity is present but zygotic activity is absent because no zygotic genome activation (ZGA) occurs. Thus, the degradation rate is significantly reduced compared with zygotic activity [7,31,59,60,61]. For maternal transcripts degradation during the development of zebrafish, a subclass of the cleared maternal mRNAs begins at fertilization, whereas others are mainly degraded after ZGA [62,63]. In *Xenopus laevis*, fertilization-induced deadenylation does not trigger decay immediately, but only after ZGA causing their deadenylation and degradation [40,64,65]. In the mouse, maternal mRNAs are degraded by both the maternal and the zygotic degradation pathways [41,66]. These are evolutionarily conserved mechanisms through which the mother provides gene products to the egg to drive the earliest stages of development.

Silkworms, like other insects such as Lepidoptera and Coleoptera, undergo superficial cleavage. In silkworm, the degradation of maternal gene mRNAs can be divided into three clusters during the MZT. Cluster 1 is stable mRNAs. In cluster 1, the mRNAs level of *Tud* is stable from zero to three hours. *Tudor* is a stress granule (SG) member that is activated upon various environmental stresses. *Tudor* (*Tud*) participates in posttranscriptional regulation in *B. mori* [67]. Silkworm *Tudor* depletion increases the levels of PIWI-interacting RNAs (piRNAs), which associate with PIWI proteins to protect genome integrity by silencing transposons in the germline [68]. Thus, cluster 1 includes stable mRNAs that perform essential housekeeping functions required during the MZT. The union of sperm and egg pronuclei occurs about two hours after silkworm eggs are laid. From about 2.0 to 2.5 h, the zygote divides repeatedly by mitosis and forms many cleavage nuclei [45,46,47]. The maternal gene mRNAs decay in clusters 2 and 3 is firstly initiated by maternal degradation activity. The initial time-point of degradation of cluster 2 is earlier than that of cluster 3. In unfertilized silkworm eggs, maternal decay activity is present, but zygotic activity is absent. Because no ZGA occurs, the degradation rate is significantly reduced in unfertilized eggs compared to that of fertilized eggs [48].

The study of transcriptional regulation has produced many discoveries that have improved our understanding of development. Understanding the post-transcriptional regulation of maternal mRNA is crucial to uncover the mechanisms that control the coordinated changes in zygotic transcription initiation [6]. The MZT represents an extreme scenario involving these mechanisms. In silkworm, according to the requirements of natural and programmed embryonic development [43,48,69], studying the establishment of transcriptional quiescence during oogenesis and identifying the first genes to be expressed during embryo (mature eggs) formation will continue to improve our understanding of transcriptional regulation during MZT.

## 4. Materials and Methods

### 4.1. Insects and Collection of Samples

*B. mori* (Dazao) larvae were reared under standard conditions (25 °C and 70% humidity). The larvae–pupae, pupae, moths, and eggs were maintained under a 12 h light/12 h dark photoperiod at 25 °C and 70% humidity. The developing oocytes (eggs) undergoing vitellogenesis, choriogenesis, and mature eggs were separately collected from ovarioles that were dissected from pharate adults eight days after larval–pupal ecdysis, according to previous studies [43,44]. The eggs of virgin moths were collected from ovarioles that were dissected from the freshly hatched female moths. Freshly hatched moths were immediately mated for 3 h, and the female moths were subsequently gathered for spawning for 15 min. Zero h is defined as the 15th minute after most female moths spawn. Then, the eggs were collected at specific points-in-time (0, 0.5, 1.0, 1.5, 2.0, 3.0, 6, 12, and 18 h) under the same conditions as previously described [48].

### 4.2. Identification of B. mori Maternal Genes

In our earlier study, we obtained 76 potential maternal genes in the *B. mori* genome [48]. We obtained their sequences from the silkworm database (SilkDB), which were used to search the NCBI database and silkBase [49]. As for the maternal genes that were similar or overlapping between the NCBI database and silkBase, the longer of the two was selected. The newly annotated protein sequences were obtained and applied to predict signal peptides by SignalP 4.1 Server.

### 4.3. Transcript Detection Reverse Transcription-PCR

Reverse transcription-PCR (RT-PCR) was used to analyze the expression patterns of maternal genes. Total RNA was extracted using TRIzol reagent (Invitrogen, Carlsbad, CA, USA) from the samples including developing oocytes (eggs) of vitellogenesis, choriogenesis, and mature eggs in day 8 pupae, the eggs of virgin moths, and after spawning at specific points-in-time (0, 0.5, 1.0, 1.5, 2.0, 3.0, 6, 12, and 18 h). Total RNA concentrations were quantified, and single-stranded cDNA was synthesized by using a PrimeScript™ RT kit (TaKaRa, Dalian, China) according to the manufacturer’s instructions. A 25 μL PCR reaction system was established by initial denaturing at 94 °C for 5 min, 35 cycles of denaturing at 94 °C for 30 s, annealing at 58 °C for 30 s, and extension at 72 °C for 30 s. This was followed by a final extension at 72 °C for 10 min before storing at 12 °C. *BmRPL3* was used as an internal control [70]. A pair of specific primers for each gene was used in RT-PCR, as shown in Appendix A. The RT-PCR product of each gene was separated by 1.2% agarose gel electrophoresis.

### 4.4. Tissue Expression Patterns Based on Microarray Database

We downloaded the microarray data from the SilkMDB to analyze tissue expression patterns of the maternal genes in 10 silkworm tissues on day 3 of the fifth instar [50]. A genome-wide microarray with 22,987 probes was designed and constructed in the silkworm genome, and each probe is also provided in this database [50]. The probes of *SPE*, *BAEE*, and *Pabn2* were not found in the database, as shown by the attached BLAST search. The microarray data of the other 63 maternal genes are provided in Appendix A. The expressed genes are defined as previously described [51]. GeneCluster 2.0 software was used to visualize the expression levels [71].

### 4.5. Transcript DecayDetection by Reverse Transcription-Quantitative PCR

Total RNA was extracted using TRIzol reagent (Invitrogen, Carlsbad, CA, USA) from the eggs collected at specific points-in-time (0, 0.5, 1.0, 1.5, 2.0, and 3.0 h). A fraction of the RNA was treated with DNase. After verifying the quality, the RNA was used to synthesize the first-strand cDNA using the PrimeScript™ RT Master Mix (Perfect Real Time; TaKaRa, Dalian, China) according to the manufacturer’s instructions. Reverse transcription-quantitative PCR (RT-qPCR) was performed as previously described [48]. A pair of specific primers for each gene was used in RT-qPCR, as shown in Appendix A.

## 5. Conclusions

In the current work, 66 maternal genes in silkworm were characterized through bioinformatics analysis and expression detection. The expression of these genes in vitellogenesis, choriogenesis, and mature eggs in day 8 pupae was analyzed using RT-PCR. We analyzed the maternal gene mRNAs decay in fertilized eggs in *B. mori* from six points-in-time by RT-qPCR. The 66 maternal genes formed three clusters of degradation patterns during the MZT. The maternal gene mRNAs of cluster 1 were stable. The initial time-point of degradation of cluster 2 was earlier than that of cluster 3. The maternal gene mRNAs decay of clusters 2 and 3 was firstly initiated by maternal degradation activity. Our findings expand upon the identification of silkworm maternal genes and provide a perspective for the embryo development in *B. mori*.

## Figures and Tables

**Figure 1 ijms-20-05651-f001:**
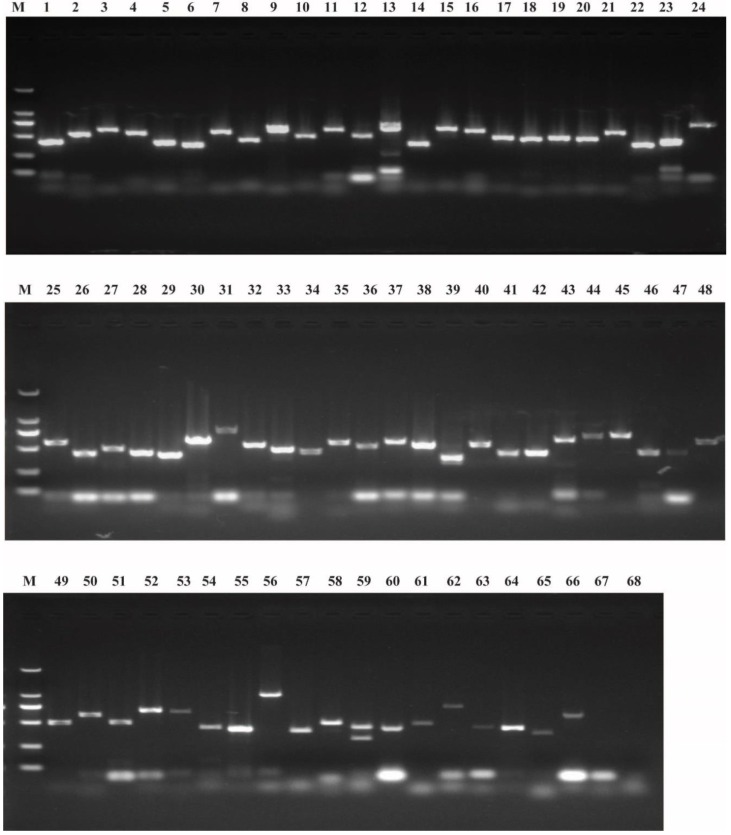
Transcriptional detection of maternal genes in eggs of virgin moths by reverse transcription (RT)-PCR. M: DL2000 DNA Maker; numbers 1 to 68 indicate *me31B*, *lok*, *vri*, *Egfr*, *Su* (*var*) *205*, *Hp1b-l*, *spz*, *tkv*, *CycB*, *proPPAE*, *asp*, *PAH*, *aub*, *Csp* (*DnaJ-7*), *SPE*, *BAEE*, *PPAE*, *Sod2*, *esc*, *Src42A*, *Smg*, *Eif-4a*, *eIF4AIII*, *rod*, *vfl*, *bai*, *Nelf-E*, *Pabn2*, *Bin1*, *tud*, *Moe*, *Sel* (*cnpy1*), *Hip14* (*ZDHHC17*), *mamo*, *sax*, *babo*, *h*, *Chc*, *Snap25*, *SPE-like*, *Src64B*, *wbl*, *Mat89Ba*, *Dif*, *ndl* (*osp*), *Nelf-A*, *tld*, *proSP7*, *gammaTub*, *Th*, *pie*, *gro*, *hb*, *pip*, *spoon* (*AKAP1*), *snk*, *Btk29A*, *dpp*, *Msp300* (*nesprin-1*), *KCNQ*, *shot*, *sog*, *Pc*, *Dst*, *TPH1*, *glo* (*hnRNPF*), BGIBMGA003296 and BGIBMGA002069, respectively.

**Figure 2 ijms-20-05651-f002:**
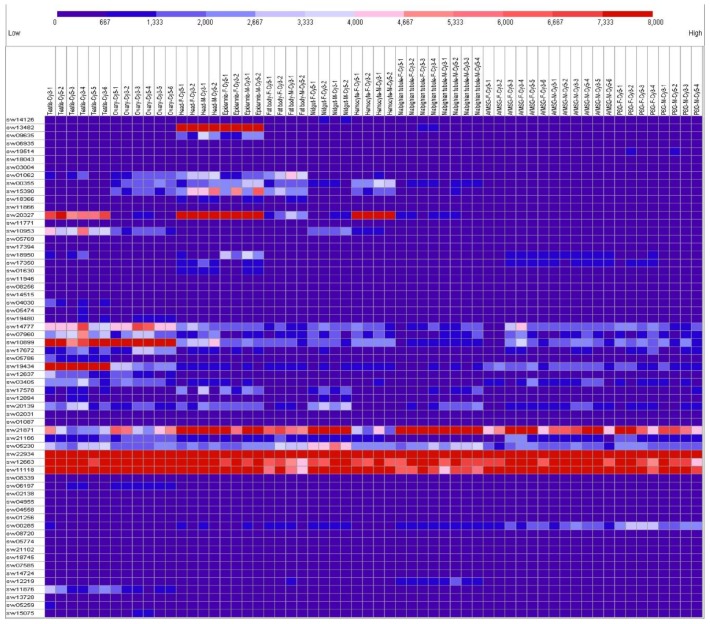
Tissue expression profile of maternal genes in larvae on day 3 of the silkworm fifth instar. The columns represent ten different tissues with both sexes: Testis, ovary, head, epidermis, fat body, midgut, hemocyte, Malpighian tubule, anterior/median silk gland (A/MSG), posterior silk gland (PSG), female (F), and male (M). Gene expression levels are represented by red (higher expression) and blue (lower expression) boxes.

**Figure 3 ijms-20-05651-f003:**
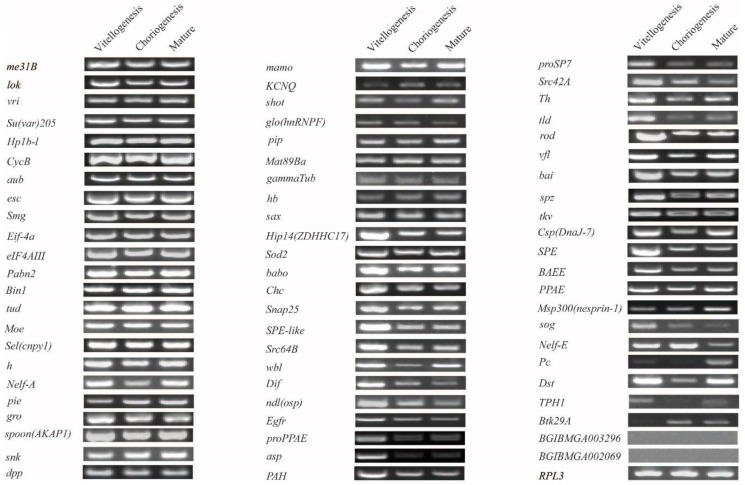
Expression patterns of maternal genes in the developing oocytes of vitellogenesis, choriogenesis, and mature eggs from day 8 female papae. Reverse transcription (RT)-PCR was performed and the *RPL3* gene was used as internal control.

**Figure 4 ijms-20-05651-f004:**
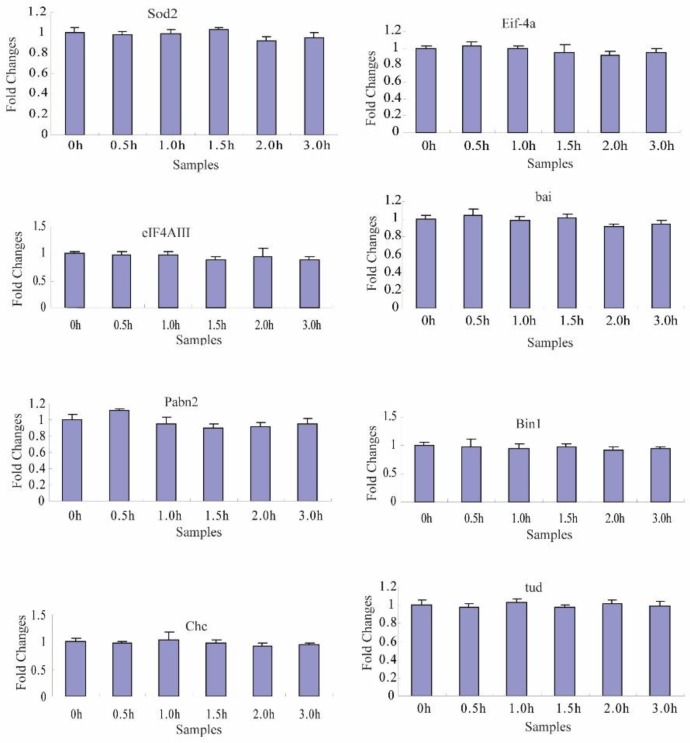
The expression profiles of cluster 1 maternal genes by RT-qPCR during the maternal-to-zygotic transition (MZT). Each time-point was replicated three times using independently collected samples. The data are the means ± SD of three independent experiments.

**Figure 5 ijms-20-05651-f005:**
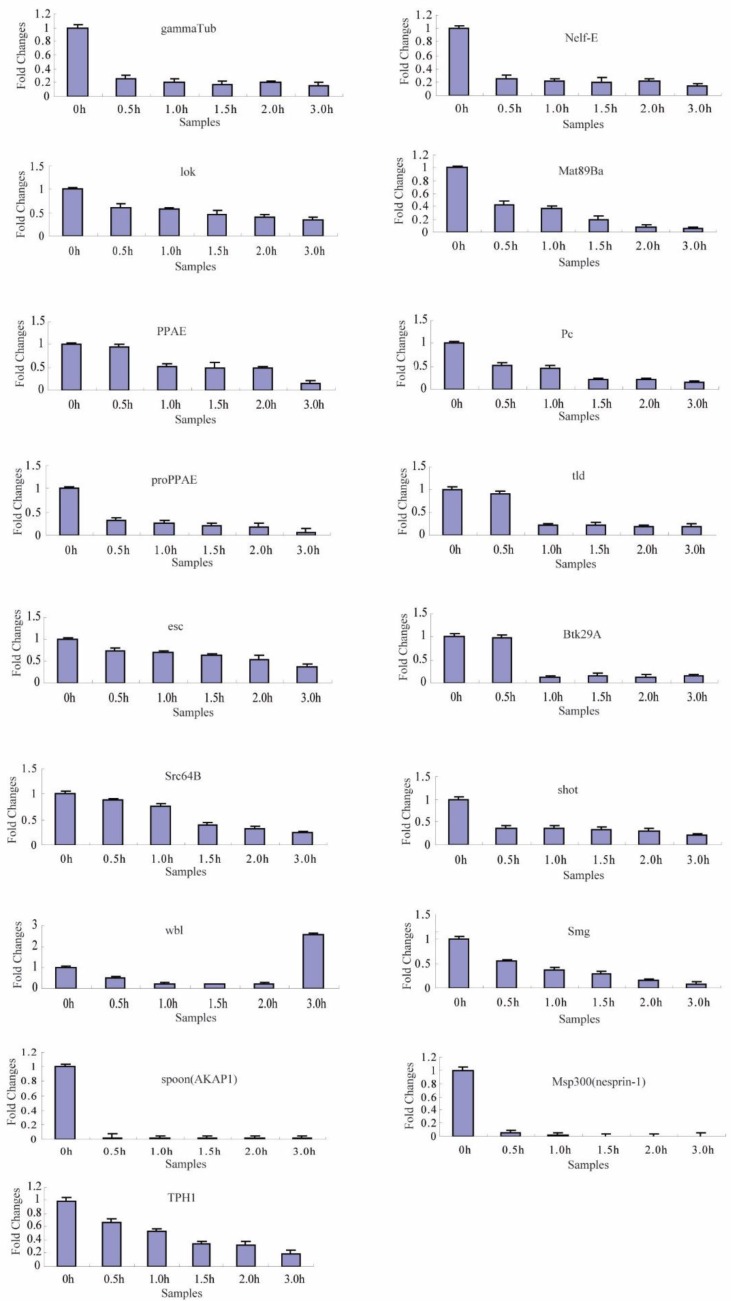
The expression profiles of cluster 2 maternal genes by RT-qPCR during the maternal-to-zygotic transition (MZT). Each time-point was replicated three times using independently collected samples. The data are the means ± SD of three independent experiments.

**Figure 6 ijms-20-05651-f006:**
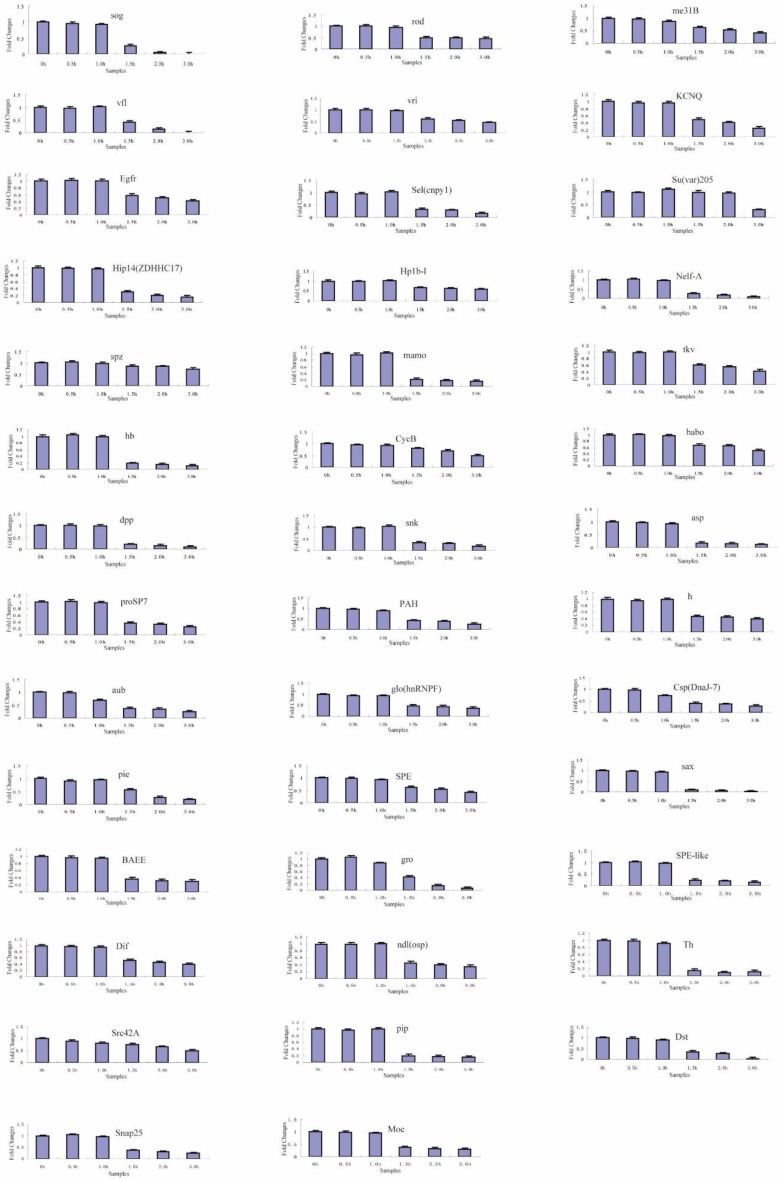
The expression profiles of cluster 3 maternal genes by RT-qPCR during the maternal-to-zygotic transition (MZT). Each time-point was replicated three times using independently collected samples. The data are the means ± SD of three independent experiments.

**Table 1 ijms-20-05651-t001:** Maternal genes in the silkworm *Bombyx mori*.

Gene Name	Accession Number	Probe	Description	Protein Length (Amino Acids)	Location (Chr.)	Signal Peptide	NCBI Reference Sequence
*me31B*	BGIBMGA010673	sw14777	ATP-dependent RNA helicase me31b	440	12	–	AK383517
*lok*	BGIBMGA005370	sw11876	cell cycle checkpoint kinase 2	528	8	–	AK382539
*vri*	BGIBMGA013421	sw05474	nuclear factor interleukin-3-regulated protein	364	27	–	AK388388
*Egfr*	BGIBMGA000602	sw11771	epidermal growth factor receptor	1449	1	1	XM_004929742
*Su* (*var*) *205*	BGIBMGA006109	sw05786	chromobox-like protein 5	191	4	–	AK385880
*Hp1b-l*	BGIBMGA012860	sw21166	heterochromatin protein	179	16	–	XM_012692600
*spz*	BGIBMGA002397	sw08256	spatzle	277	9	1	NM_001114594
*tkv*	BGIBMGA007355	sw21102	bone morphogenetic protein receptor type-1B	485	3	–	AK385287
*CycB*	BGIBMGA003747	sw04030	cyclin B homolog	525	5	1	AK382330
*proPPAE*	BGIBMGA013746	sw20327	prophenoloxidase activating enzyme precursor	441	28	1	AK383056
*asp*	BGIBMGA005594	sw10953	protein abnormal spindle	2309	17	–	XM_004921876
*PAH*	BGIBMGA003866	sw01062	phenylalanine hydroxylase	456	1	–	NM_001287837
*aub*	BGIBMGA010644	sw10899	aubergine protein	899	12	–	EU143547
*Csp* (*DnaJ-7*)	BGIBMGA007808	sw01087	dnaJ (Hsp40) homolog 7	203	15	–	XM_012692267
*SPE*	BGIBMGA005172	–	serine protease easter	430	25	1	XM_012689474
*BAEE*	BGIBMGA005173	–	BzArgOEtase	369	25	1	NM_001043379
*PPAE*	BGIBMGA010546	sw15390	prophenoloxidase activating enzyme	382	12	1	AK383498
*Sod2*	BGIBMGA007453	sw21871	Mn superoxide dismutase	221	3	–	XM_012690443
*esc*	BGIBMGA006325	sw12637	extra sex combs	411	6	–	AK385410
*Src42A*	BGIBMGA004089	sw11866	tyrosine-protein kinase Src42A-like	922	19	–	XM_012693691
*Smg*	BGIBMGA008249	sw17394	*Bombyx mori* protein Smaug	599	18	–	AK385418
*Eif-4a*	BGIBMGA003186	sw22934	eukaryotic translation initiation factor 4A	420	4	–	AK383662
*eIF4AIII*	BGIBMGA004822	sw12663	eukaryotic initiation factor 4A-III	405	25	–	AK386335
*rod*	BGIBMGA002655	sw06197	rough deal protein	1817	28	–	XM_004932260
*vfl*	BGIBMGA012283	sw05259	zinc finger protein	1064	1	–	XM_004933146
*bai*	BGIBMGA004891	sw00285	transmembrane trafficking protein	205	25	–	AK385774
*Nelf-E*	BGIBMGA003207	sw19434	negative elongation factor E	264	13	–	AK385219
*Pabn2*	BGIBMGA001950	–	polyadenylate binding protein 2	225	19	–	XM_012696483
*Bin1*	BGIBMGA011014	sw11118	histone deacetylase complex subunit SAP18	159	23	–	AK384481
*tud*	BGIBMGA011857	sw17672	maternal protein tudor	1839	11	–	XM_012695006
*Moe*	BGIBMGA002544	sw02031	moesin/ezrin/radixin homolog 1	574	9	–	AK383231
*Sel* (*cnpy1*)	BGIBMGA003267	sw18745	protein canopy homolog 1	242	2	–	AK385660
*Hip14* (*ZDHHC17*)	BGIBMGA001083	sw14724	palmitoyltransferase ZDHHC17	591	13	–	XM_004927675
*mamo*	BGIBMGA012517	sw18043	zinc finger protein	798	9	–	XM_012688563
*sax*	BGIBMGA009134	sw06935	activin receptor type-1	566	20	1	XM_004925975
*babo*	BGIBMGA000601	sw20139	TGF-beta receptor type-1	503	1	–	XM_012693543
*h*	BGIBMGA005390	sw08720	protein hairy isoform	261	8	–	XM_004932202
*Chc*	BGIBMGA012935	sw07960	*Bombyx mori* clathrin heavy chain	1681	16	–	AK378376
*Snap25*	BGIBMGA005176	sw12219	synaptosomal-associated protein 25	211	25	–	AK383225
*SPE-like*	BGIBMGA013797	sw18366	serine protease easter-like	431	28	1	AK386026
*Src64B*	BGIBMGA012094	sw07585	tyrosine-protein kinase Src64B	521	11	–	AK378283
*wbl*	BGIBMGA012931	sw05230	*Bombyx mori* protein windbeutel	254	16	1	AK381984
*Mat89Ba*	BGIBMGA007162	sw18950	nucleolar protein 6	1120	21	–	AK385389
*Dif*	BGIBMGA010496	sw17578	embryonic polarity protein dorsal isoform	529	12	–	AK386522
*ndl* (*osp*)	BGIBMGA014089	sw15075	ovarian serine protease	1920	9	–	XM_012691651
*Nelf-A*	BGIBMGA002236	sw03405	negative elongation factor A	581	26	–	XM_012691503
*tld*	BGIBMGA002518	sw11946	tolloid-like protein 1	1349	9	1	XM_012694771
*proSP7*	BGIBMGA012427	sw09635	serine protease 7 precursor	397	21	1	AK386200
*gammaTub*	BGIBMGA013500	sw02138	tubulin gamma-1	456	15	–	AK377270
*Th*	BGIBMGA000563	sw13482	tyrosine hydroxylase	561	1	–	AK383721
*pie*	BGIBMGA001789	sw19480	G2/M phase-specific E3 ubiquitin-protein ligase	757	11	–	XM_004922174
*gro*	BGIBMGA012449	sw19514	groucho-like isoform X1	679	21	–	AK382427
*hb*	BGIBMGA003334	sw12894	protein hunchback	621	15	–	AK385224
*pip*	BGIBMGA011817	sw14126	heparan sulfate 2-O-sulfotransferase pipe	436	11	–	XM_004931477
*spoon* (*AKAP1*)	BGIBMGA006841	sw04955	A-kinase anchor protein 1	3601	10	–	XM_004924760
*snk*	BGIBMGA001745	sw01630	venom protease-like	401	11	1	XM_004922131
*Btk29A*	BGIBMGA000972	sw08339	tyrosine-protein kinase Btk29A	610	13	–	XM_012691697
*dpp*	BGIBMGA010384	sw00355	decapentaplegic	369	12	1	XM_012693077
*Msp300* (*nesprin-1*)	BGIBMGA010471	sw17350	nesprin-1	8514	12	–	XM_012693124
*KCNQ*	BGIBMGA003731	sw13728	potassium voltage-gated channel subfamily KQT member 5	751	9	–	XM_012693718
*shot*	BGIBMGA004414	sw05774	*Bombyx mori* plectin-like	1325	20	–	XM_012695224
*sog*	BGIBMGA005348	sw05769	dorsal-ventral patterning protein Sog	927	8	1	XM_012695533
*Pc*	BGIBMGA006904	sw14515	polycomb	281	10	–	AK383962
*Dst*	BGIBMGA004415	sw01256	*Bombyx mori* dystonin-like	4811	20	–	XM_012695102
*TPH1*	BGIBMGA000642	sw03004	tryptophan 5-hydroxylase 1	543	1	–	NM_001309589
*glo* (*hnRNPF*)	BGIBMGA013473	sw04558	heterogeneous nuclear ribonucleoprotein F	336	6	–	XM_012690736

“–” indicates that no signal peptide was predicted and no probe number was found.

**Table 2 ijms-20-05651-t002:** The characteristics of maternal genes mRNA decay.

Cluster	No. of Maternal Genes	Name of Maternal Genes
1	8	*Sod2*, *Pabn2*, *Eif-4a*, *Bin1*, *eIF4AIII*, *Chc*, *bai*, *tud*
2	17	*gammaTub*, *Nelf-E*, *lok*, *Mat89Ba*, *PPAE*, *Pc*, *proPPAE*, *tld*, *esc*, *Btk29A*, *Src64B*, *shot*, *wbl*, *Smg*, *spoon*(*AKAP1*), *Msp300*(*nesprin-1*), *TPH1*
3	41	*sog*, *rod*, *me31B*, *vfl*, *vri*, *KCNQ*, *Egfr*, *Sel*(*cnpy1*), *Su*(*var*)*205*, *Hip14*(*ZDHHC17*), *Hp1b-l*, *Nelf-A*, *spz*, *mamo*, *tkv*, *hb*, *CycB*, *babo*, *dpp*, *snk*, *asp*, *proSP7*, *PAH*, *h*, *aub*, *glo*(*hnRNPF*), *Csp*(*DnaJ-7*), *pie*, *SPE*, *sax*, *BAEE*, *gro*, *SPE-like*, *Dif*, *ndl*(*osp*), *Th*, *Src42A*, *pip*, *Dst*, *Snap25*, *Moe*

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
