# Peer review of "Expression Analysis of mRNA Decay of Maternal Genes during Bombyx mori Maternal-to-Zygotic Transition"

_ijms, 2019, doi:10.3390/ijms20225651_

Round 1
Reviewer 1 Report
In this article titled “The expression analysis of mRNA decay of maternal genes during Bombyx mori maternal-to-zygotic transition” authors describe by RT-qPCR the expression of 66 genes during the early development of the silkworm (from 0.5 to 3.0 hours after spawning). Moreover, by end point RT-PCR, they study the expression of the same genes in eggs of virgin moths and in developing oocytes (during vitellogenesis and choriogenesis and in mature eggs. Also, they study the expression of 8 genes belonging to cluster 1, in “eggs across different developmental stages”.
The results could be interesting but they are not well presented and described, especially because the manuscript is written in a very bad English and is hard to read, and also because the introduction does not contains enough background.
Authors should allocate time for revising the manuscript completely.
Therefore, first of all, I suggest to revise English carefully. Some sentences are not comprehensible.
Pay attention to punctuation and spelling.
Moreover, stages and hours of sample collection should be better indicated.
Importantly, revise completely the end point RT-PCR results and discussions, since end point PCRs are not quantitative experiments.
The Discussion section should include: Summary of your results and their interpretation in light of known literature, Importance of your results, Shortcomings of the study and Future directions. The background necessary to comprehend should be included in the Introduction section.
102 “The probes of SPE, BAEE and Pabn2 are not found in SilkMDB”. Which kind of probe are they? Please better explain what you are doing. Material and methods are too beyond.
Please describe accurately data downloaded. For example: “sw20327 (proPPAE) was higher in the ovary, testis...”. sw20327 is expressed in testis but virtually not in ovary. Therefore, check results.
109 “These results may suggest that the transcripts of silkworm maternal genes are produced during the development of oocytes in late pupae”. What does this sentence mean?
118 “2.4. Expression Analysis in Developing Oocytes in Day 8 Pupae”, and also results shown in Fig5. Authors describe and discuss their results as RT-PCR was a quantitative experiment. RT-PCR cannot be used to actually quantitatively determine the amount of gene expression (at least at the end point).
154 “Unlike other genes, the transcript of wbl gene was decreased significantly from 0.5 h to 2.0 h, and increased sharply at 3.0 h. This transcript belongs to cluster 2 and represent an exception”.
After correction and indication of the exceptionality, the sentence can be moved after the sentence “This indicates that maternally supplied mRNAs of most maternal genes were universally degraded during the maternal-to-zygotic transition”.
181-188 “The embryonic development of B. mori is significantly different from Drosophila. According to variously morphological criteria, the development of the follicles was divided into 12 different stages [28]. During vitellogenesis (stage4-10), the oocyte increases gradually in volume and is gradually filled with yolk spheres, lipid droplets, and glycogen granules, moreover at the end of this period degenerated nurse cells are devoured by follicular epithelial cells [28]. In choriogenesis period (stage11-12) which is mainly the formation of egg-shell structure with different types of egg-shell proteins being synthesized and secreted successively to construct the egg-shell, and then the eggs maturation [27, 28]”.
This background could be more useful in the introduction. Please, revise it and move it.
Figure 8 should be moved after discussion. In the discussion the reason(s) why and how authors distinguish between “maternal degradation pathway” and “maternal and zygotic degradation pathways” should be explained carefully. Moreover, the maternal-to-zygotic transition is a universal process in animal development, when the embryo activates zygotic gene expression and thus no longer solely relies on maternally provided transcripts. the elimination of many maternal mRNAs coincides with the activation of zygotic transcription. Therefore, it should be remembered that authors cannot distinguish between maternal products and zygotic products of same or similar genes.
259 4.5. Transcript Decay Detection by qPCR à 4.5. Transcript Decay Detection by RT-qPCR
(Reverse transcription and quantitative PCR)
Check titles of references
Author Response
Response to Reviewer 1 Comments
Point 1: The results could be interesting but they are not well presented and described, especially because the manuscript is written in a very bad English and is hard to read, and also because the introduction does not contains enough background.
Response 1: Thanks for the reviewer’s good evaluation and kind suggestion. Due to your suggestion, (ⅰ): our manuscript has been revised in MDPI for English editing by selecting specialist editing (English editing ID: English-13162); (ⅱ): the section of "Introduction" has also been supplemented, adjusted and improved. The revisions are highlighted using the "Red Font" and "Track Changes" function in Microsoft Word.
Point 2: Authors should allocate time for revising the manuscript completely. Therefore, first of all, I suggest to revise English carefully. Some sentences are not comprehensible. Pay attention to punctuation and spelling.
Response 2: Thanks for the reviewer’s kind suggestion. We carefully revised our manuscript including sentence smoothness, punctuation and spelling, etc, and also sent the manuscript to MDPI for English revision. The revisions are highlighted using the "Red Font" and "Track Changes" function in Microsoft Word.
Point 3: Moreover, stages and hours of sample collection should be better indicated.
Response 3: Thanks for the reviewer’s kind suggestion. We have improved the stages and hours of sample collection that can be found in Line 288-298 (page 9) in "Materials and Methods".
Point 4: Importantly, revise completely the end point RT-PCR results and discussions, since end point PCRs are not quantitative experiments.
Response 4: Thanks for the reviewer’s good evaluation. Due to your evaluation, in the revision, (ⅰ): we have already distinguished RT-PCR and RT-qPCR in terms of their full representation; (ⅱ): we have better indicated RT-PCR and RT-qPCR in the sections where they appear; (ⅲ): in order to clearly indicate the section of "Transcriptional Degradation during the Maternal-to-Zygotic Transition", the Figure 5 (Cluster 1 maternal genes expression patterns in eggs across different developmental stages) has been adjusted to Figure S1presented in the "Supplementary Methods". The order of other images has also been accordingly changed.
Point 5: The Discussion section should include: Summary of your results and their interpretation in light of known literature, Importance of your results, Shortcomings of the study and Future directions. The background necessary to comprehend should be included in the Introduction section.
Response 5: Thanks for the reviewer’s good evaluation and kind suggestion. The corresponding revisions are highlighted using the "Red Font" in the Discussion and Introduction sections. The advices from you have provided with great helps to us in the current work and will improve our level of scientific research in the future work. Thank you very much.
Point 6: 102 “The probes of SPE, BAEE and Pabn2 are not found in SilkMDB”. Which kind of probe are they? Please better explain what you are doing. Material and methods are too beyond.
Response 6: Thanks for the reviewer’s kind suggestion. We have added the details in Line 139-141, page 2 using the "Red Font" in the revision. In the year of 2007, Xia et al designed and constructed a genome-wide microarray with 22,987 70-mer oligonucleotides of each probe covering the presently known and predicted genes in the silkworm genome. In 2019, a high-quality genome assembly of silkworm has been accomplished and is provided in SilkBase (http://silkbase.ab.a.u-tokyo.ac.jp). The new genome assembly and gene models reflected more accurate coding sequences and gene sets than old ones. In current work, we search the SilkBase of new genome assembly, 70-mer oligonucleotides of SPE, BAEE, and Pabn2 designed and constructed depending on the old gene models, which doesn't match by a BLAST search. The microarray data of the other 63 maternal genes were provided in Table S2 in the revision.
Point 7: Please describe accurately data downloaded. For example: “sw20327 (proPPAE) was higher in the ovary, testis...”. sw20327 is expressed in testis but virtually not in ovary. Therefore, check results.
Response 7: Thanks for the reviewer’s good evaluation and kind suggestion. After carefully checking the results, we found that the description of sw20327 (proPPAE) and sw14126 (pip) was wrong in the previous manuscript. We are sorry. The description of virtual expression is "The expression level of sw20327 (proPPAE) was higher in the testis, head, epidermis, and hemocyte, and sw13482 (Th) was higher in the head and epidermis" in the revision. Thank you very much for your careful reviews in our work.
Point 8: 109 “These results may suggest that the transcripts of silkworm maternal genes are produced during the development of oocytes in late pupae”. What does this sentence mean?
Response 8: Thanks for the reviewer’s good evaluation. Our description of this sentence that is not rigorous and accurate. We have replaced this sentence with "Most maternal genes showed low expression levels in multiple silkworm larval tissues on day 3 of the fifth instar; this is contrary to the abundant expression in the embryo (Figure 1)" in the revision.
Point 9: 118 “2.4. Expression Analysis in Developing Oocytes in Day 8 Pupae”, and also results shown in Fig5. Authors describe and discuss their results as RT-PCR was a quantitative experiment. RT-PCR cannot be used to actually quantitatively determine the amount of gene expression (at least at the end point).
Response 9: Thanks for the reviewer’s good evaluation. In the revision, we have already distinguished RT-PCR and RT-qPCR in terms of their full representation and have better indicated them in the sections where they appear. The Figure 5 (Cluster 1 maternal genes expression patterns in eggs across different developmental stages) has been adjusted to Figure S1 provided in the "Supplementary Methods". The order of other images has also been accordingly changed.
Point 10: 154 “Unlike other genes, the transcript of wbl gene was decreased significantly from 0.5 h to 2.0 h, and increased sharply at 3.0 h. This transcript belongs to cluster 2 and represent an exception”. After correction and indication of the exceptionality, the sentence can be moved after the sentence “This indicates that maternally supplied mRNAs of most maternal genes were universally degraded during the maternal-to-zygotic transition”.
Response 10: Thanks for the reviewer’s good evaluation and kind suggestion. According to your suggestion, we have made the adjustment in the revision. Thank you very much for your careful reviews in our work.
Point 11: 181-188 “The embryonic development of B. mori is significantly different from Drosophila. According to variously morphological criteria, the development of the follicles was divided into 12 different stages [28]. During vitellogenesis (stage4-10), the oocyte increases gradually in volume and is gradually filled with yolk spheres, lipid droplets, and glycogen granules, moreover at the end of this period degenerated nurse cells are devoured by follicular epithelial cells [28]. In choriogenesis period (stage11-12) which is mainly the formation of egg-shell structure with different types of egg-shell proteins being synthesized and secreted successively to construct the egg-shell, and then the eggs maturation [27, 28]”.
This background could be more useful in the introduction. Please, revise it and move it.
Response 11: Thanks for the reviewer’s good evaluation and kind suggestion. According to your suggestion, we have moved it to the Introduction in Line 80-86, page 2. New cohesive contents are prepared and can be found in Line 226-233, page 8.
Point 12: Figure 8 should be moved after discussion. In the discussion the reason(s) why and how authors distinguish between “maternal degradation pathway” and “maternal and zygotic degradation pathways” should be explained carefully. Moreover, the maternal-to-zygotic transition is a universal process in animal development, when the embryo activates zygotic gene expression and thus no longer solely relies on maternally provided transcripts. The elimination of many maternal mRNAs coincides with the activation of zygotic transcription. Therefore, it should be remembered that authors cannot distinguish between maternal products and zygotic products of same or similar genes.
Response 12: Thanks for the reviewer’s good evaluation and kind suggestion. In the revision, Figure 8 has been removed after discussion and the related contents have also been revised. Your good evaluations and kind suggestions improve our understanding of the mechanisms of maternal genes degradation. Thank you very much.
Point 13: 259 4.5. Transcript Decay Detection by qPCR à 4.5. Transcript Decay Detection by RT-qPCR
(Reverse transcription and quantitative PCR)
Response 13: Thanks for the reviewer’s good evaluation. The related content has been revised.
Point 14: Check titles of references
Response 14: Thanks for the reviewer’s kind suggestion. The titles of references have been carefully checked in the revision. Thank you very much for your careful reviews in our work.
Reviewer 2 Report
In this manuscript, authors identified 66 maternal genes in Bombyx mori using bioinformatics and expression detection during maternal-to-zygotic transition. Moreover, expression profiles will help understanding for the functions of these maternal genes. Authors provided abundant data in the manuscript. However, the language need be improved. Some grammatical mistakes need be revised carefully in the manuscript. And, the “Materials and Methods” need more detail.
Author Response
Response to Reviewer 2 Comments
Point 1: In this manuscript, authors identified 66 maternal genes in Bombyx mori using bioinformatics and expression detection during maternal-to-zygotic transition. Moreover, expression profiles will help understanding for the functions of these maternal genes. Authors provided abundant data in the manuscript. However, the language need be improved. Some grammatical mistakes need be revised carefully in the manuscript. And, the “Materials and Methods” need more detail.
Response 1: Thanks for the reviewer’s good evaluation and kind suggestion. Due to your suggestion, (ⅰ): we carefully revised our manuscript including sentence smoothness, punctuation and spelling, etc, and also sent the manuscript to MDPI for English revision editing by selecting specialist editing (English editing ID: English-13162); (ⅱ): the section of “Materials and Methods” has also been supplemented, adjusted and improved. The revisions are highlighted using the "Red Font" and "Track Changes" function in Microsoft Word. Thank you very much for your careful reviews in our work.
Round 2
Reviewer 1 Report
The revised version has been improved significantly and authors have done a good job to revise the paper.
However the manuscript needs an additional reading and revision.
The important conceptual correction that have to be done regards the RT-PCR results. Only quantitative PCR (in real time) can give us quantitative results, therefore authors cannot describe PCR results in terms of more or less expression (or RNA levels).
All the suggestions are depicted in the notes of the PDF file attached.

Author Response
Response to Reviewer 1 Comments
Point 1: However the manuscript needs an additional reading and revision. All the suggestions are depicted in the notes of the PDF file attached.
Response 1: Thanks for the reviewer’s good evaluation and kind suggestion. All the suggestions are depicted in the notes of the PDF file that have been revised one to one. The adjustments are highlighted and can be found in Line 21-24, 33, 75, 82, 88-92, 144, 157, 171-181, 201-205, 219, 234-236, 270-272, and 289 in the revision. Thank you very much for your careful and precise reviews in our work.
Point 2: The important conceptual correction that have to be done regards the RT-PCR results. Only quantitative PCR (in real time) can give us quantitative results, therefore authors cannot describe PCR results in terms of more or less expression (or RNA levels).
Response 2: Thanks for the reviewer’s good evaluation and kind suggestion. Now we know the differences between RT-PCR and RT-qPCR. The descriptions of RT-PCR results are virtual corrections that can be found in Line 171-181, Page 3 and Line 201-205, Page 4. The careful and precise reviews, and the strategic and constructive advices from you have provided with great helps to us in the current work and will improve our level of scientific research in the future work. In our future work on silkworm maternal genes, we have obtained the inspirations and ideas from your strategic and constructive suggestions. Thank you very much.